# Oncological Treatment Adverse Reaction Prediction: Development and Initial Validation of a Pharmacogenetic Model in Non-Small-Cell Lung Cancer Patients

**DOI:** 10.3390/genes16030265

**Published:** 2025-02-24

**Authors:** Concetta Cafiero, Raffaele Palmirotta, Canio Martinelli, Alessandra Micera, Luciano Giacò, Federica Persiani, Andrea Morrione, Cosimo Pastore, Claudia Nisi, Gabriella Modoni, Teresa Galeano, Tiziana Guarino, Ilaria Foggetti, Cecilia Nisticò, Antonio Giordano, Salvatore Pisconti

**Affiliations:** 1Medical Oncology, SG Moscati Hospital, 74010 Statte, Italy; concettacafiero@gmail.com (C.C.); cosimo.pastore@asl.taranto.it (C.P.); claudia.nisi@asl.taranto.it (C.N.); gabriella.modoni@asl.taranto.it (G.M.); teresa.galeano@asl.taranto.it (T.G.); tiziana.guarino@asl.taranto.it (T.G.); ilafog@gmail.com (I.F.); salvatorepisconti@hotmail.com (S.P.); 2Anatomic Pathology Unit, Fabrizio Spaziani Hospital, 03100 Frosinone, Italy; 3Interdisciplinary Department of Medicine, School of Medicine, University of Bari “Aldo Moro”, 70124 Bari, Italy; 4Sbarro Institute for Cancer Research and Molecular Medicine and Center for Biotechnology, Department of Biology College of Science and Technology, Temple University, Philadelphia, PA 19122, USA; canio.martinelli@temple.edu (C.M.); andrea.morrione@temple.edu (A.M.); antonio.giordano@temple.edu (A.G.); 5Gynecology and Obstetrics Unit, Department of Human Pathology “G. Barresi”, University of Messina, 98125 Messina, Italy; 6Research and Development Laboratory for Biochemical, Molecular and Cellular Applications in Ophthalmological Sciences, IRCCS-Fondazione Bietti, 00184 Rome, Italy; micera.alessandra@gmail.com; 7Bioinformatics Core Facility, Gemelli Science and Technology Park (G-STeP), Fondazione Policlinico Universitario A. Gemelli IRCCS, 00168 Rome, Italy; luciano.giaco@policlinicogemelli.it (L.G.); federica.persiani@guest.policlinicogemelli.it (F.P.); 8Medical Oncology Unit, ASL Frosinone, 03100 Frosinone, Italy; cecilia.nistico@aslfrosinone.it; 9Department of Medical Biotechnology, University of Siena, 53100 Siena, Italy

**Keywords:** pharmacogenomics, individual variability, supportive care, pharmacogenetics, cancer, oncology, symptom management, personalized medicine

## Abstract

**Background/Objectives:** The accurate prediction of adverse drug reactions (ADRs) to oncological treatments still poses a clinical challenge. Chemotherapy is usually selected based on clinical trials that do not consider patient variability in ADR risk. Consequently, many patients undergo multiple treatments to find the appropriate medication or dosage, enhancing ADR risks and increasing the chance of discontinuing therapy. We first aimed to develop a pharmacogenetic model for predicting chemotherapy-induced ADRs in cancer patients (the ANTIBLASTIC DRUG MULTIPANEL PLATFORM) and then to assess its feasibility and validate this model in patients with non-small-cell lung cancer (NSCLC) undergoing oncological treatments. **Methods:** Seventy NSCLC patients of all stages that needed oncological treatment at our facility were enrolled, reflecting the typical population served by our institution, based on geographic and demographic characteristics. Treatments followed existing guidelines, and patients were continuously monitored for adverse reactions. We developed and used a multipanel platform based on 326 SNPs that we identified as strongly associated with response to cancer treatments. Subsequently, a network-based algorithm to link these SNPs to molecular and biological functions, as well as efficacy and adverse reactions to oncological treatments, was used. **Results:** Data and blood samples were collected from 70 NSCLC patients. A bioinformatic analysis of all identified SNPs highlighted five clusters of patients based on variant aggregations and the associated genes, suggesting potential susceptibility to treatment-related toxicity. We assessed the feasibility of the platform and technically validated it by comparing NSCLC patients undergoing the same course of treatment with or without ADRs against the cluster combination. An odds ratio analysis confirmed the correlation between cluster allocation and increased ADR risk, indicating specific treatment susceptibilities. **Conclusions:** The ANTIBLASTIC DRUG MULTIPANEL PLATFORM was easily applicable and able to predict ADRs in NSCLC patients undergoing oncological treatments. The application of this novel predictive model could significantly reduce adverse drug reactions and improve the rate of chemotherapy completion, enhancing patient outcomes and quality of life. Its potential for broader prescription management suggests significant treatment improvements in cancer patients.

## 1. Introduction

Advances in cancer therapy, including immunotherapy, chemotherapy, targeted agents, hormonal therapy, surgery, and radiation therapy, have significantly improved cancer treatment and patient outcomes. However, most of these treatments can lead to a wide range of toxicities, which can significantly impact patients’ quality of life and long-term health [1].

Adverse drug reactions (ADRs) are often underestimated, although they are the cause of approximately 10% of all hospital admissions in Europe [2]. ADRs are strongly associated with treatment response, clinical outcome, and individual quality of life, and they are associated with skin reactions (mucosal damage), as well as cardiological, hematological, and visual (ocular surface discomfort, dry eye) system impairments and even fatigue [2]. ADRs are still very poorly characterized, even though they constitute a limitation for any therapy, especially for patients with comorbidities [3].

This stark reality underscores the critical importance of minimizing treatment-related toxicities in oncological care. Generally, individual variability in response to drugs is due to multiple factors, including i. physiology (age, sex, body weight, physical condition); ii. pathology (diseases, liver or kidney function, visual impairments); iii. environmental (diet, alcohol, tobacco, other drugs) factors, and iv. genetic (gene polymorphisms) factors [4].

Regarding this last point, it is well established that ADRs have a strong genetic predisposition [3,5], and a vast number of studies have demonstrated that numerous polymorphic variants of genes responsible for the metabolism and action of antiblastic drugs are involved in the toxicity associated with oncological treatments [1,6]. Patients can respond differently to the same drug, and these differences are linked to many proteins involved in drug therapy response [6].

However, despite numerous studies on this topic to date, most chemotherapeutic or antiblastic drugs, with only few exceptions, are dosed solely on the basis of individual body surface area, which is related to circulating blood volume and glomerular filtration [7,8].

Consequently, the focus of ongoing research should shift towards the development of more efficient and safer strategies to predict and mitigate the risk of ADRs for individual patients, with the aim of enhancing patient compliance and overall treatment success.

Given their potential impact, it is crucial to emphasize the importance of identifying gene variants involved in the metabolism and action of anticancer drugs. Understanding these variations can significantly influence the choice of personalized treatment, minimize drug toxicities, and better predict responses to chemotherapy and/or radiotherapy [9].

In this new setting, the pharmacogenetics of chemotherapeutic drugs would offer clinicians the chance to know the functional polymorphisms of enzymes involved in the metabolism of the most used chemotherapy drugs alone or in combination with radiotherapy. It would represent a reliable and easy-to-use way of limiting toxicity and predicting the efficacy of a therapeutic scheme, in order to adapt the dosage and allow the personalization of genetic-based therapies (precision medicine or individualized therapy) [10,11].

The ability to discriminate between fit and unfit patients for a specific oncological treatment based on genetic background would be ideal, and in fact there are studies that prospect decision-making algorithms based on calculating the probability of developing an ADR to a specific drug by analyzing genetic profiles [12].

However, at the moment there are no “easily interpreted” multigene analysis systems able to predict the toxicity and/or efficacy of pharmacological treatments of neoplastic diseases. Existing systems only analyze specific gene variants and do not provide a comprehensive panel that simultaneously assesses both the pharmacokinetics of enzymes and the impact of genetic variants on drugs responses. Essentially, while we can analyze how some drugs interact with certain genetic markers, we lack a unified system that integrates all these factors to provide a complete assessment of drug response and toxicity [10,13].

In this paper, we present an innovative, user-friendly hybrid platform aimed at minimizing ADRs for individual patients undergoing specific oncological treatments, thereby optimizing their therapeutic regimen. We discuss technical, developmental, and validation data in the context of the latest advancements in pharmacogenomics.

## 2. Materials and Methods

### 2.1. Study Design and Participants

This study describes the development and patenting of a novel pharmacogenetic predictive model (the ANTIBLASTIC DRUG MULTIPANEL PLATFORM), followed by its feasibility assessment and validation through a prospective observational single-cohort study named “Method for determining targeted drug therapy in malignant lung cancer through a personalized antiblastic drug multipanel platform”. Unlike typical predictive systems, which are derived directly from training data, our model was constructed using well-established datasets available in the literature.

To develop the ANTIBLASTIC DRUG MULTIPANEL PLATFORM, we focused on functional polymorphisms, particularly those well documented in the scientific literature for their associations with cancer biology and chemotherapy mechanisms. We initially identified potential candidate SNPs using the Pharmacogenomics Knowledgebase (Pharmgkb.org) website database (https://www.pharmgkb.org/ accessed on 15 May 2021). It is an online, publicly available knowledge database that includes detailed information regarding the impact of human genetic variation on drug response [14]. We identified 326 SNPs, using a stepwise approach. First, we utilized the “Variant Annotations” function within the Pharmacogenomics Knowledgebase website (Appendix A*)*. This function essentially compiles published studies linking SNPs to specific drug effects. This allowed us to systematically identify SNPs potentially associated with drug toxicity. We then used the “Clinical Annotations” function, which provides practical guidance on these SNP–drug associations based on the underlying data. This function was particularly helpful in identifying SNPs linked to severe side effects (“Toxicity”) associated with each drug. Finally, to ensure the accuracy and clinical relevance of our findings, the identified SNPs were further examined and validated using the 1000 Genomes Browsers (A Deep Catalog of Human Genetic Variation, https://www.internationalgenome.org accessed on 1 June 2021), Genome Aggregation Database v3.1 (GnomAD, https://gnomad.broadinstitute.org accessed on 4 June 2021), and Single Nucleotide Polymorphism Database (dbSNP, http://www.ncbi.nlm.nih.gov/snp/ accessed on 6 June 2021) [15,16,17]. This step was essential to confirm the reliability of our data within the broader context of pharmacogenomics.

For the prospective observational single-cohort study titled “Method for determining targeted drug therapy in malignant lung cancer through a personalized antiblastic drug multipanel platform”, all eligible patients had a diagnosis of NSCLC from stage T2 to T4 and from N1 to N3, with an age between 40 and 75 years and with a PS ECOG of not more than 2. All patients were candidates for oncological treatments such as single or multiple chemotherapy or concurrent radio chemotherapy. Exclusion criteria were as follows: ECOG Performance Status over 2, severe comorbidities such as severe cardiovascular diseases, multiorgan pathologies, immunological diseases that forbad immunotherapy, early-stage NSCLC patient candidates to locoregional treatments such as surgery, and stereotactic radiosurgery as recommended by guidelines referring to the AIOM (Associazione Italiana Oncologia Medica—Italian association of medical oncology). This study was conducted at the Medical Oncology Department of the S.G. Moscati Hospital, Taranto, Italy. At first assessment of the enrolled patients, we performed venous blood sampling using an EDTA vacutainer as part of our clinical activities and extracted germinal DNA from an aliquot of peripheral blood (3–4 mL), coded and stored at a controlled temperature (−20 °C). Then, we analyzed DNA with next-generation molecular analysis techniques following standardized protocols. We assigned to the samples and each enrolled patient a progressive numerical code. Patients started the oncological treatments following existing guidelines, including single-agent chemotherapy (cisplatin or carboplatin), combined chemotherapy (carboplatin/pemetrexed), radiotherapy, or concomitant chemoradiotherapy with platinum compounds. We monitored the patients during therapy to assess all the ADRs experienced during treatment.

All clinical, preanalytical, and analytical procedures were conducted according to the principles expressed in the Declaration of Helsinki and the Guideline for Good Clinical and Laboratory Practice. The observational study was approved by the Ethics Committee of the ASL Brindisi (Italy) (R.CE. 183/20) on 4 December 2020. The study did not influence the therapeutic approach or drug lineage, and all procedures were performed in accordance with approved procedures for oncological treatments. Informed consent was signed by the NSCLC patients for the collection of clinical data, blood samples, and DNA analysis.

We followed the TRIPOD (Transparent Reporting of a multivariable prediction model for Individual Prognosis Or Diagnosis) checklist for reporting the study of the development and validation of the ANTIBLASTIC DRUG MULTIPANEL PLATFORM.

### 2.2. Model Development

The ANTIBLASTIC DRUG MULTIPANEL PLATFORM, which we have developed and patented, allows the analysis of germinal DNA extracted from solid, liquid, pre-extracted biological samples, biopsy tissues, peripheral blood, buccal mucosa, impression cytology, or any fluid containing inflammatory cells, all biological matrices from which constitutional DNA can be extracted.

The analytical method relies on the use of an adaptable technology on any type of old or new-generation molecular platform capable of DNA sequencing/mutational analysis (Next-Generation Sequencing, Array, Nanostring, etc.) even from existing amplified processed products and/or those stored in dedicated biobanks. As described above, the 326 SNP polymorphic variants included in the ANTIBLASTIC DRUG MULTIPANEL were identified among genes responsible for drug metabolism, transport, absorption, distribution, and excretion, as well as in those genes coding for drug receptor proteins. Variants found in patients are then stratified and matched with “clinical annotations” using an integrated bioinformatics analysis, to determine the appropriate pharmacological response and instrumental therapy, such as radiotherapy, for the specific cancer patient studied. Based on this analysis, the physician can then assess oncological patients and guide them towards personalized/targeted therapies, choosing drug(s) and routes of administration before starting the therapeutic protocol. In addition, based on the information provided by the method and in case of limited therapeutic options, the clinician could decide on the use of a potentially toxic drug in the absence of a therapeutic alternative, by appropriately modulating the dosage.

In 2021, Cafiero and Pisconti obtained the Italian patent “Innovative method for therapeutic addressing in the oncological field” (Patent number Deposit # U1020 102019000024385 of 15 December 2021, classification C12Q) for the ANTIBLASTIC DRUG MULTIPANEL PLATFORM. The patent also provides an application (or App), which can be downloaded onto a mobile device (Android/iOS) and allows an easy to interpret visualization.

The App generates a report (digital/protected) of the results based on gene variants (SNPs) found in the patient under investigation, so that the physician can choose the targeted cancer therapy. The final report is unique and up to date on all gene variants found to date, as in fact it is updated in real time with the list of variants found in any tumor type and antiblastic agent provided by PharmGKB. It is important to highlight that this patented method was designed to adapt, allowing for updates in the detection of SNPs.

### 2.3. Model Feasibility Assessment and Validation

To assess the feasibility and clinical applicability of the ANTIBLASTIC DRUG MULTIPANEL PLATFORM, we conducted a clinical trial titled “Method for determining targeted drug therapy in malignant lung cancer through a personalized antiblastic drug multipanel platform”. This trial was carried out from January 2020 to July 2023 at the Medical Oncology Department of S.G. Moscati Hospital in Taranto, Italy. We applied the platform to 70 patients with NSCLC, predicting potential side effects associated with the oncological treatments selected and based on each patient’s specific pathological condition, prior to the initiation of their guideline-based therapy. A detailed methodology is provided below (i). Throughout the treatment, patients were monitored for side effects.

(i)Sample Preparation, Next-Generation Sequencing, and validation analysis

The 326 SNPs of interest on which the platform is based were submitted via web interface for primer pool design and synthesis using the proprietary Ion Ampliseq Designer algorithm (Ion AmpliSeq Designer version 7.6.4; Thermo Fisher Scientific, Waltham, MA, USA). Genomic DNA was isolated from peripheral blood using the DNeasy^®^ blood and tissue kit (QIAGEN Inc., Chatsworth, CA, USA) and quantified with a Qubit^®^ 3.0 fluorometer (Life Technologies™, Carlsbad, CA, USA), according to the manufacturer’s protocol. DNA extracts (10 ng/sample) were amplified and barcoded using Ion AmpliSeq Library Kit 2.0 and Ion Xpress barcode adapters (Thermo Fisher Scientific, Waltham, MA, USA). The library was purified with AMPure XP Reagent (Beckman Coulter Inc., Brea, CA, USA) and quantified with the Ion Library Quantitation Kit (Life Technologies, Thermo Fisher Scientific) on the StepOne Plus system (Applied Biosystem, Thermo Fisher Scientific). Template preparation was performed with an Ion Chef™ instrument using the Ion 520™ & Ion 530™ Kit-Chef. Sequencing of the amplicon libraries was carried out on a 530 chip using the Ion Torrent S5 system (Thermo Fisher Scientific) according to the supplier’s instructions. Sequence results were analyzed using Torrent Suite Software 5.0.5, and all reads were aligned to the human reference hg19 genome. Variant calling was performed with the Torrent Variant Caller plugin version 5.0.4.0, while Ion Reporter Software v.5.6 (Thermo Fisher Scientific) was used for downloading the VCF files and the evaluation of single nucleotide variants.

(ii)Data Analysis

Once the VCF files were obtained, subsequent bioinformatic analyses allowed the identification of specific clusters for groups of patients in relation to genetic variants and related genes. This protocol is schematically illustrated in Figure 1, while a detailed description of its methods is provided in Appendix A.

### 2.4. Statistical Analysis

The clinical database was used to produce a spreadsheet (xls file format) reporting the number of targeted therapy cancer patients and the dichotomous outcome of patients with or without ADRs in respect to cluster association. MedCalc ver. 22.021 for Windows 11 64-bit (MedCalcSoftwareLtd. © 2024) was used for odds ratios (ORs) and Chi-square calculations. Results are indicated in the text as OR values with low and high limits according to 95% confidence intervals (CIs). As a rule, ORs less than 1 favor this cluster model. The graphics were created using the GraphPad Prism 9.4 software (GraphPad Software; San Diego, CA, USA).

## 3. Results

We designed an SNP panel as described above (Figure 1) and then performed an NGS evaluation of 70 germinal DNA extracts from venous blood samples. Out of the 326 initially identified SNPs, 309 (94.7%) were successfully sequenced and subjected to subsequent computational analysis. In the design process, 15 SNPs were not included by the algorithm, and 2 SNPs were excluded from the subsequent analysis as they turned out to be duplicates of the same SNP. Single-nucleotide variants (SNVs) were the predominant alterations, with a single case of INDELs (insertions and deletions), and were all distributed across autosomal chromosomes, with the exception of a single SNP on the X chromosome. Regarding the performance of analysis in the NGS, as reported by Ion Reporter™ Software, the samples had an average read depth >500× and a Phred quality score (Q score), representing the probability that a given base is called incorrectly by the sequencer, of 30 (*p*: 0.001).

The panel included both polymorphisms for which the guidelines recommend their detection during therapy, such as *DPYD*, *UGT1A*, *CYP2C19*, *TMPT*, and *SLCO1B1*, and validated variants reported in the literature as particularly useful for therapeutic decisions, such as *MTHFR*, *TYMS*, *ERCC1*, *XRCC1*, *GSTP1*, *CYP3A4/3A5,* or *ABCB1.* In this regard, it is possible to access pharmacogenetic information from several drug regulatory agencies, including the FDA and EMA, on the Pharmgkb website (https://www.pharmgkb.org/labelAnnotations accessed on 10 February 2025). In the panel, there were also 15 variants related to radiotherapy-related toxicity events (Appendix A).

All SNPs, except five located in non-coding regions of the genome, were distributed within 176 genes classified into different categories: transporters, phase I metabolism enzymes, phase II metabolism enzymes, DNA repair genes, and a group including drug targets, cell signaling, transcription, cell cycle, apoptosis, and other related genes (Appendix A).

In our initial bioinformatic analysis, we examined the mutational profiles of patients and discovered that patients segregated into five clusters based on variant aggregation and the affected genes. Analyses were repeated several times with identical results. These clusters were statistically significant, as indicated by a modularity score of Q = 0.06. In spite of a low modularity, stemming from strong patient similarity, five distinct clusters were identified. The clusters, numbered 0 to 4, varied in size and included 21 patients (30%), 16 patients (22.8%), 27 patients (38.5%), 5 patients (7.1%), and 1 patient (1.4%), respectively (Appendix A).

Our analysis revealed a distinctive genetic landscape: 146 genes were exclusively mapped within a single cluster, while 30 genes were shared by no more than three clusters. The distribution of the unique genes across clusters 0 to 4 was as follows: 47, 37, 45, 12, and 5, respectively. Notably, no genes or their variants were common across all five clusters, as illustrated in Figure 2A. This Venn diagram displays specific genes and their associated variants within each cluster. Cluster 4 was unique, as in fact it was represented by a single patient and characterized by specific gene variants not shared with other clusters. Figure 2B shows additional cluster specifics, showing the number of patients, genes, and variants for each specific cluster. It is also evident that clusters 0, 1, and 2 have a higher number of patients, genes, and variants as compared to clusters 3 and 4.

After obtaining the list of genes from this population via NGS, we conducted a Gene Ontology (GO) enrichment analysis. This analysis classified the genes into specific categories related to their “Biological Process” and “Molecular Function” within each cluster of patients, as detailed in Appendix A.

Each patient cluster is characterized by unique combinations of variants, where each variant is associated with distinct biological processes and molecular functions. This analysis demonstrated that each cluster identifies patients with uniquely defined biological activities and functional capabilities.

Finally, we utilized data from PharmGKB that details the associations between variants and their well-known related drugs. This allowed us to link each cluster with specific drug impacts, enhancing our understanding of personalized treatment options. Since a drug can be annotated to more than one variant, we calculated the proportion of each drug present in a given cluster relative to its total occurrences in the dataset, as illustrated in the bar plot in Figure 3.

Further analysis, depicted in the Venn diagram in Appendix A, revealed a specific distribution of drugs across the clusters: cluster 0 has 3 specific drugs, cluster 1 has 14, cluster 2 has 7, and cluster 3 has 2, while cluster 4 does not possess any exclusive drug. Additionally, 41 drugs are shared among the clusters, with 12 of these common to all five clusters. Significantly, cluster 4 was associated only with 12 drugs that are shared by all clusters, as detailed in Figure 3, Appendix A.

### Cluster–ADR Association Analysis

The clinicopathological characteristics of the 70 NSCLC patients included in this study are summarized in Table 1, and distribution according to cancer stages and cluster is summarized in Appendix A.

To evaluate whether our clustering approach could predict a patient’s susceptibility (based on their pharmacogenetic profile) to ADRs following specific oncological treatments, we tested the likelihood that a cluster could predict an ADR when treatment was administered, comparing it to the probability of an ADR occurring without treatment.

Thus, we used MedCalc software for the OR calculation (OR values with upper and lower limits calculated with a 95% CI), starting from clusters and ADR outcomes for each treatment. An odds ratio equal to 1 (the same as a probability of 0.5 or 95% CI) means that exposure does not affect the odds of the outcome, an odds > 1 implies that exposure is associated with higher odds of an outcome, and an OR < 1 means that exposure is associated with lower odds of an outcome.

The validity of the results obtained with this approach was tested by evaluating and comparing the OR values with the percentages of variants associated with response to specific therapies distributed across each cluster (Figure 3, Appendix A).

Using this approach, we observed that in the case of radiotherapy, cluster 0, presenting 75% of variants related to this treatment, shows the majority of positive associations (OR, 2.40; 0.30–19.40), in contrast to cluster 1 (OR, 1.50; 0.18–12.46) and cluster 2 (OR, 0.86; 0.09–8.07), which show a percentage of variants of 8.3% and 16.6%, respectively (Figure 4A).

For patients receiving cisplatin therapy, cluster 0 displayed the highest positive association (OR, 5.25; 0.49–56.80), with 32.2% of variants compared to cluster 1 (OR, 0.31; 0.02–56.80) and cluster 2 (OR, 0.88; 0.06–12.98), both with 27.4% of variants (Figure 4B).

Regarding data from carboplatin therapy, our analysis showed a high OR value for cluster 1 (OR, 4.58; 0.46–45.61) and not for cluster 0 (OR, 1.64; 0.32–8.45), although we expected different results given the higher frequency of variants (39.6%) in cluster 0 compared to cluster 1 (19%) (Figure 4C). A possible explanation can be that a large proportion of these patients in cluster 1 were also treated with pemetrexed, for which the frequency of variants is higher (40%) in cluster 1 than in cluster 0 and cluster 2 (both 20%). Indeed, the analysis of pemetrexed treatment shows a high OR for cluster 1 (OR, 8.33; 0.63 –110.03) compared to cluster 0 (OR, 0.33; 0.03–3.2) and cluster 2 (OR, 0.67; 0.06–6.87) (Figure 4D), and the results obtained by aggregating carboplatin and pemetrexed data strongly support our explanation (Figure 4E).

In our analysis, we also aggregated data obtained after treatment with carboplatin and cisplatin (platinum compounds) to generate a more statistically relevant sample size. Again, the data indicated a higher association for cluster 0 (OR, 2.22; 0.59–8.26) than clusters 1 (OR, 1.20; 0.30–4.74) and cluster 2 (OR, 0.50; 0.14–1.79) (Figure 4F). Then, we evaluated the association with radiotherapy in combination with platinum compounds (cisplatin or carboplatinum) and this analysis showed that a stronger association was related again to cluster 0 and equally to cluster 1 (OR, 4.00; 0.27–60.33) and was lower in cluster 2 (OR, 1.20; 0.06–24.47) (Figure 4G).

Since cluster 0 resulted mostly in an OR >1, we can conclude that in our model cluster 0 showed an increased occurrence of ADR events (risk response or positive association) (Figure 4).

## 4. Discussion

In this study, we developed and validated a pharmacogenetic model to predict the ADRs associated with oncological treatments such as chemotherapy (single or multiple drugs), radiotherapy, and concurrent chemoradiotherapy. The ADRs associated with oncological treatments are a severe problem for cancer patients, as in fact ADRs affect a patient’s quality of life [18], often causing permanent damage to vital organs like the heart, lungs, liver, kidneys, and eyes [19] and leading to long-term cognitive and peripheral nervous system impairments [16]. The reduction in quality of life is also associated with chronic issues such as fatigue, pain, and ocular discomfort [20]. Late diagnoses and suboptimal drug selection, primarily due to ADRs, undermine the efficacy of treatment strategies [21]. The ability to predict therapeutic management would effectively minimize ADRs [22].

Data in the literature support the use of pharmacogenomics (PGx) to improve treatments by tailoring therapeutic strategies based on individual genetic profiles, thereby minimizing toxicity, optimizing drug efficacy [23], and reducing healthcare costs [24]. Genomic variants can significantly affect patient responses by reducing ADRs [25], with most individuals carrying actionable PGx alleles that are pivotal for drug–gene interactions [26,27]. Thus, an in-depth characterization of individual genomic profiles is the best approach toward precision medicine in oncology [28].

However, the complexity of molecular data often makes PGx results challenging to interpret, thereby complicating their clinical applications [29]. In addition, the utility of PGx is limited by the requirement for accessible and quick interpretation tools that do not delay treatment [30].

Our platform addresses these issues by incorporating a comprehensive dataset of 326 SNPs, simplifying decision-making processes for clinicians and enabling faster, more effective treatment adjustments. These 326 SNPs were selected for their key roles in pharmacogenomics. This approach is based on the use of a single sampling of venous blood and a specific genomic assay to evaluate related cancer biomarkers for the best therapeutic management [31]. In this way, we can not only predict individual drug responses but also link genetic profiles to potential ADRs, as supported by the literature, since these ADRs are commonly associated with certain drugs.

We assessed its feasibility and validated this platform on a cohort of 70 NSCLC patients coming from Puglia (the southern region of Italy), confirming both its non-invasive nature for patients and user-friendly features for clinicians. In this specific analysis, 326 SNPs were initially identified, 309 (94.7%) of which were successfully sequenced and computationally analyzed. The Ion Ampliseq Designer did not include 17 SNPs (as explained in the development section), but this exclusion does not compromise the applicability of the platform for this subset of cancer patients, and it does not result in missing clinical information, as in fact two of these SNPs were synonymous, and the remaining 15 were considered irrelevant for the drugs used in treating lung cancer, as reported in 2019 when the patent application was filed (see Appendix A). It is important to mention that this patented tool is designed to accommodate updates in detectable SNPs according to evolving scientific data and can be applied to patients undergoing various kind of treatments. After the genetic analysis, we correlated SNPs with molecular and biological functions using a network-based algorithm, and stratified patients into five distinct clusters. Each cluster was able to predict different drug responses. The platform simplified the interpretation of results for clinicians, allowing them to make informed decisions without analyzing each SNP individually. The easy inclusion of individual patients in a given cluster, with immediate evidence of the greater or lesser likelihood that a therapy could potentially be associated with toxicity, was one of the most interesting pieces of data that emerged from the results after SNP analysis of this specific population and subsequent GO enrichment analysis (Figure 3, Appendix A). This is because the platform’s analysis is based on 309 SNPs per individual. This generates a unique patient-specific genetic profile that is linked to a certain risk of ADRs, making it inherently personalized.

The stratification of a very specific subset of patients into five clusters revealed a significant insight: each cluster demonstrates a strong and positive correlation between specific SNPs and ADRs associated with a specific oncological treatment. This finding underscores the potential of genetic profiling to predict treatment response and tailor therapy to individual patients.

Thus, variations in patient genetic profiles can lead to different clustering outcomes, demonstrating the platform’s flexibility and precision in tailoring results to a specific genetic background. Since our approach includes SNPs related to common cancer therapies, it can be also applied, after appropriate bioinformatic evaluation, to other cancer models. Furthermore, we can foresee the extension of this hybrid approach to other non-oncological pathologies such as neurological, metabolic, endocrinological or even ophthalmological diseases, with applications in clinical and emergency settings.

Our goal was focused on the technical and methodological validation of this patented hybrid platform rather than a clinical validation, for which additional patient cohorts and larger numbers are definitely needed [32]. Obviously, clinical validation requires a different methodological approach, not simply correlational and observational but interventional. In other words, a validation study in which the patient’s treatment is expected to be changed and modified according to the prospective prediction of the response to adverse drug reactions based on the genotype identified in the correlative phase.

In this way, the ANTIBLASTIC DRUG MULTIPANEL PLATFORM could significantly improve therapy outcomes by integrating the screening of specific genetic variants associated with selected drugs.

Of course, we would like to point out that there are other tools that are based on an association of variants or changes in certain genes that are able to make similar predictions.

Our system is based on a germline assessment of SNPs reported and evaluated by Pharmgkb to provide a prediction of potential personal susceptibility to ADRs.

An interesting recent study takes, instead, an interestingly different approach, using single-cell RNA-seq (scRNA-seq) technology and datasets such as The Cancer Genome Atlas (TCGA) to obtain neoplasm gene expression data that correlate with drug resistance predictions [33]. The system turns out to be very effective and also applicable to different types of cancer. This suggests that a combination of approaches is likely to be the way forward for the future of oncology pharmacogenetics.

By tailoring treatment to each patient’s genetic profile, this tool could potentially increase survival rates and decrease both the side effects experienced by patients and costs for the healthcare system [34]. Tailoring anticancer drugs based on disease-associated biomarkers and drug-specific genetic variants could also enhance the success rates of therapies [27].

Moreover, the ability to rank gene variants and provide results in an easily interpretable format to oncologists represents a significant advancement in the clinical application of pharmacogenomics and personalized medicine [35,36,37].

## 5. Conclusions

The urgent need to tailor oncological treatments to minimize ADRs is well recognized. With advancements in pharmacogenomics and data analysis, our platform offers a significant step forward. By integrating genetic data with the efficacy of oncological treatments and their associated ADRs, this platform predicts patient responses in a personalized manner and presents these insights intuitively to clinicians. This approach not only aims to reduce ADRs, enhancing treatment compliance and patient quality of life, but it also has the potential to increase the overall survival rate and reduce healthcare costs. Moreover, as modern medicine shifts towards viewing diseases as multifactorial issues that require an omics-based approach, our model integrates genomic, pharmacogenomic, and radiogenomic data [38,39,40]. This integration makes it a powerful tool for personalized medicine and opens new doors to integrated omics, positioning it as a key solution in the evolution of healthcare. The findings coming from the clinical assessment and validation of our system suggest the use in the near future of this tool to assist oncologists in selecting drugs tailored for a specific patient [41]. This novel model tool may represent one of the first real applications of precision medicine and individualized therapy.

## Figures and Tables

**Figure 1 genes-16-00265-f001:**
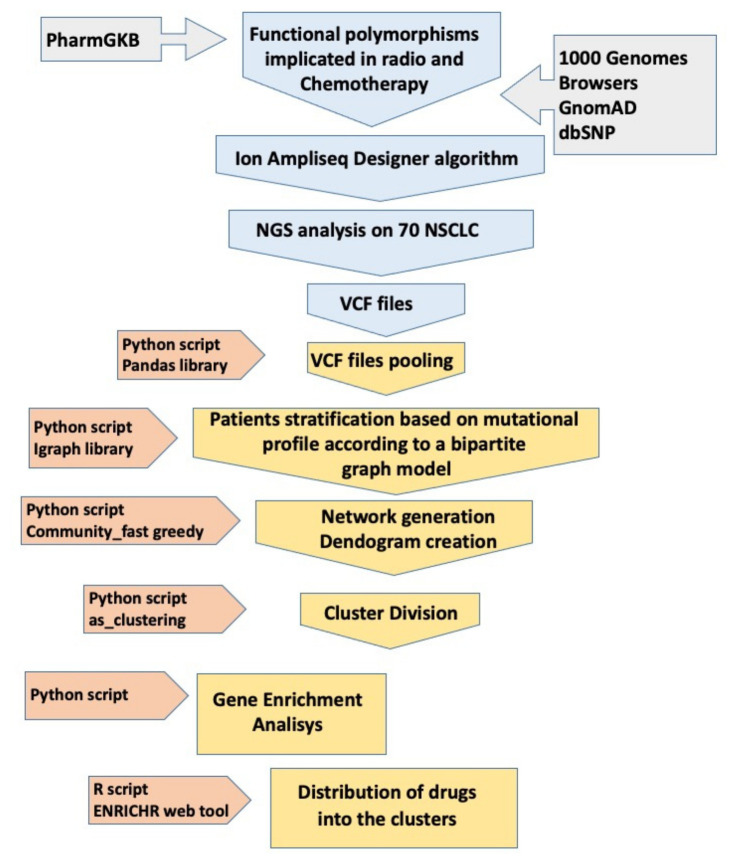
Flowchart of the pipeline of the protocol used for the study.

**Figure 2 genes-16-00265-f002:**
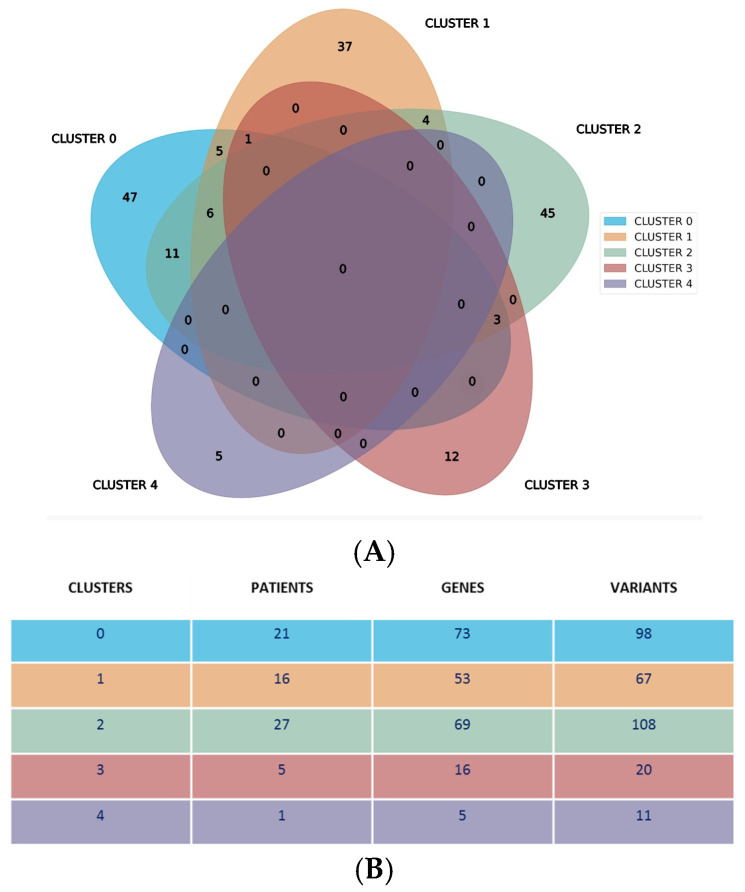
Specifics of identified clusters. (**A**) Distribution of various genes related to the specific variants for each cluster. (**B**) Number of patients, genes, and variants identified in each cluster.

**Figure 3 genes-16-00265-f003:**
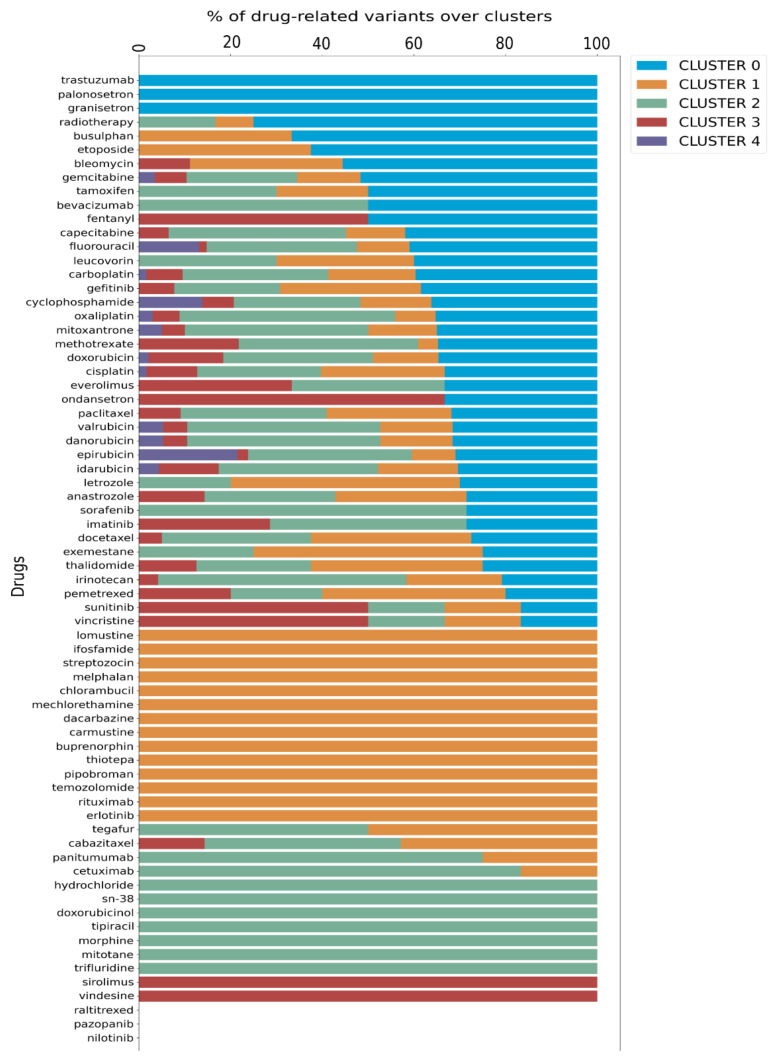
Distribution of drugs in the 5 clusters identified based on variants observed. The fraction of each drug is associated with each cluster with respect to the total presence in the dataset.

**Figure 4 genes-16-00265-f004:**
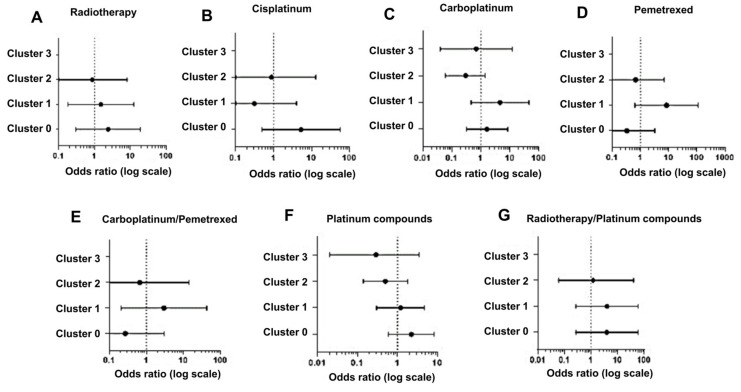
Graphical representation of odds ratio with 95% confidence interval. The association between clusters and therapy is shown.

**Table 1 genes-16-00265-t001:** Clinical and oncological characteristics of the cohort of NSCLC patients (N = 70). SD: Standard Deviation.

Clinical Variables	
Mean age ± SD	69.5 ± 7.8
*Gender, N (%)*	
Male	49 (70%)
Female	21 (30%)
*Cancer stages*	
Stage IA	6 (8.57%)
Stage IB	7 (10%)
Stage IIA	11 (15.7%)
Stage IIB	7 (10%)
Stage IIIA	13 (18.57%)
Stage IIIB	6 (8.57%)
Stage IV	20 (28.57%)
*Therapy toxicity, N (%)*	
Patients with ADRs	37 (52.8%)
Hematologic toxicity	14 (20%)
Gastrointestinal toxicity	7 (10%)
Asthenia	13 (18.5%)
Skin toxicity	2 (2.8%)
Pain symptoms	8 (11.4%)
Dyspnea	5 (7.1%)
*Therapy, N (%)*	
Chemotherapy regimen	67 (95.7%)
Platinum-based doublet chemotherapy	44 (62.8%)
Cisplatin	14 (20%)
Carboplatin	30 (42.8%)
Non-platinum	26 (37.1%)
Radiotherapy	17 (24.2%)

## Data Availability

The datasets generated during and/or analyzed during the current study are available from the corresponding author on reasonable request.

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
