# Peer review of "Oncological Treatment Adverse Reaction Prediction: Development and Initial Validation of a Pharmacogenetic Model in Non-Small-Cell Lung Cancer Patients"

_genes, 2025, doi:10.3390/genes16030265_

Round 1
Reviewer 1 Report
Comments and Suggestions for Authors
Oncological Treatment Adverse Reaction Prediction: development and initial validation of a Pharmacogenetic Model in Non-Small Cell Lung Cancer Patients. by Concetta Cafiero et al. is a well written manuscript describing a study using a multigene panel to predict response to anticancer therapies in NSCLC patients. This is an important topic of research and more data, “user friendly” panels, and algorithms are needed to move the field of cancer PGx forward.
Comments:
The authors explain that the panel includes not only PK genes, but also PD genes (drug receptors), but then also state that essentially any DNA sample from the patient could be used. For PK genes this makes sense, but for PD genes, wouldn’t tumor DNA have to be used? Are the PD variants related to mutations required for targeted drug response? Looking at the list of genes/variants it seems no, but this should be clarified.
Looking at eTable 2 there may be some errors e.g. DPYD is listed under drug targets, but is a phase 1 metabolic enzyme and is important for 5-FU metabolism. This table should be thoroughly checked for any mistakes.
In the second paragraph under the results section, the authors indicate that there were some genes tested “for which guidelines are mandatory in their detection during therapy”. Can the authors indicate which guidelines these are? In the US, guidelines may recommend, but “mandate” would be too strong of a description. I understand it may differ where the study took place. Some clarification or comparison to US and other system guidelines may be helpful.
While this is a step forward in Cancer PGx and personalized therapy, because the predictions are based on clusters, it is possible that someone may not fit into one of the clusters that were derived from this population. Can the authors discuss how this can be overcome or future options to improve their model?
The panel the authors designed could be used for many different medications than are typically used for NSCLC. Can the authors explain why they chose NSCLC patients as their only population for this study?
In the last paragraph of the discussion, the authors state that their platform “can markedly enhance therapy outcomes” but this is only theoretically, since there was no cohort in which there was genotype guided dosing to compare to. This is an important next step of validating the model and should be clarified. In addition, the authors may want to discuss use of and/or need for guidelines to optimize dosing/medication choice if toxicities are anticipated based on genotyping.
Author Response
Response to Reviewer 1
Oncological Treatment Adverse Reaction Prediction: development and initial validation of a Pharmacogenetic Model in Non-Small Cell Lung Cancer Patients. by Concetta Cafiero et al. is a well written manuscript describing a study using a multigene panel to predict response to anticancer therapies in NSCLC patients. This is an important topic of research and more data, “user friendly” panels, and algorithms are needed to move the field of cancer PGx forward.
Comments:
The authors explain that the panel includes not only PK genes, but also PD genes (drug receptors), but then also state that essentially any DNA sample from the patient could be used. For PK genes this makes sense, but for PD genes, wouldn’t tumor DNA have to be used? Are the PD variants related to mutations required for targeted drug response? Looking at the list of genes/variants it seems no, but this should be clarified.
Authors’ reply: We are grateful to the Reviewer for critically revising our work and giving appropriate suggestions, certainly aimed at the manuscript improvement. First of all, we apologize if we have been unclear in the text of the paper and have taken some concepts for granted.
SNPs, known to be germline polymorphic sequence variants with a certain allelic frequency in the general population, were evaluated for both pharmacodynamic (PD) and pharmacokinetic (PK) genes. The analysis of these polymorphisms was always performed on the germinal (constitutional) and not on somatic DNA level, thus not on nucleic acids from tumor.
Effectively, in the part of the paper to which the reviewer refers, we have argued that our method "allows analysis of DNA extracted from solid, liquid, pre-extracted biological samples, biopsied tissues, peripheral blood, buccal mucosa, cytology imprint or any fluid containing inflammatory cells". Then, following the reviewer's criticism, we implemented the text and made the concept more specific.
Regarding the last question, all SNPs (we prefer this term rather than variants) analyzed in our study were selected from the database of the Pharmacogenomics Knowledgebase platform (Pharmgkb.org) website (https://www.pharmgkb.org/) as well as detailed in Materials and Methods (2.1. Study design and participants). The PharmGKB is an NIH-funded database of human genetic variation affects response to medications, clinically actionable gene-drug associations and genotype-phenotype relationships. On the other hand, a detailed list of SNPs used and drug susceptibility, faithfully reported for each SNP by Pharmgkb, is listed in Supplementary Table 1.
Looking at eTable 2 there may be some errors e.g. DPYD is listed under drug targets, but is a phase 1 metabolic enzyme and is important for 5-FU metabolism. This table should be thoroughly checked for any mistakes.
Authors’ reply: Following the Reviewer’s suggestion, the eTable 2 has been corrected.
In the second paragraph under the results section, the authors indicate that there were some genes tested “for which guidelines are mandatory in their detection during therapy”. Can the authors indicate which guidelines these are? In the US, guidelines may recommend, but “mandate” would be too strong of a description. I understand it may differ where the study took place. Some clarification or comparison to US and other system guidelines may be helpful.
Authors’ reply: We fully agree with the reviewer. The term mandatory is not accurate. The text has been amended as suggested by the reviewer, and a cross-reference has also been added to the interesting WEB page (https://www.pharmgkb.org/labelAnnotations) on the Pharmgkb website, where it is possible to evaluate the recommendations of various drug regulatory agencies, including the FDA, EMA, HCSC, Swissmedic and PMDA.
While this is a step forward in Cancer PGx and personalized therapy, because the predictions are based on clusters, it is possible that someone may not fit into one of the clusters that were derived from this population. Can the authors discuss how this can be overcome or future options to improve their model?
Authors’ reply: Of course, we do not consider the clusters identified in this initial study on a pilot population to be exhaustive or complete. On the other hand, we also specified in the Discussion that “variations in patient genetic profiles can lead to different clustering outcomes, demonstrating the platform's flexibility and precision in tailoring to specific genetic background”.
We assume that every patient analysed with this system must be fully examined according to the flow chart described in Materials and Methods. In detail, each patient must undergo next-generation sequencing (NGS). The resulting data must then be analysed using bioinformatic tools to provide specific output in relation to genetic variants. Finally, these results will be represented in tabular and graphical form, illustrating susceptibility to specific drugs (see Figure 3). Therefore, it is possible that by extending the study to a larger population, additional specific clusters will be identified.
The panel the authors designed could be used for many different medications than are typically used for NSCLC. Can the authors explain why they chose NSCLC patients as their only population for this study?
Authors’ reply: We agree with the reviewer and even argue that “Since our approach includes SNPs related to common cancer therapies, it can be also applied, after appropriate bioinformatic evaluation, to other cancer models” (Discussion, page 11). The choice to analyse this neoplasm is purely practical, since for an initial validation it was necessary to have at least a homogeneous population of a certain size in order to proceed with the study. Another reason, no less important, was the limited budget allocated to the study (during the COVID period), which allowed neither an increase in the number of samples nor an extension of the analyses to other types of neoplasia.
In the last paragraph of the discussion, the authors state that their platform “can markedly enhance therapy outcomes” but this is only theoretically, since there was no cohort in which there was genotype guided dosing to compare to. This is an important next step of validating the model and should be clarified. In addition, the authors may want to discuss use of and/or need for guidelines to optimize dosing/medication choice if toxicities are anticipated based on genotyping.
Authors’ reply: We agree with the reviewer in fact argue that “Our goal was focused on the technical and methodological validation of this patented hybrid platform rather than a clinical validation, for which additional patient cohorts and larger numbers are definitely needed”. Obviously, clinical validation requires a different methodological approach, not simply correlational and observational, but interventional. In other words, a validation study in which the patient's treatment is expected to be changed and modified according to the prospective prediction of the response to adverse drug reactions based on the genotype identified in the correlative phase. The text has been modified as suggested by the reviewer.
Reviewer 2 Report
Comments and Suggestions for Authors
· PS ECOG under 3 should explain in more detail what this means in order to facilitate monitoring and understanding of respondents' status
· „Every patient needs to be fit for oncological treatment with a PS ECOG under 3. Exclusion criteria were: ECOG Performance Status over 2“ – this part should be clarified more precisely, because if they are more than 2 as an exclusion factor, and in the previous sentence it is under 3
· The authors should clarify how they did it: "We constructed the ANTIBLASTIC DRUG MULTIPANEL PLATFORM?
· Table 1 - the table should not be split into two pages, what is DS? Mean (SD) ? The authors should explain this with a short explanation below the table
· Figure 4 - make the diagrams uniform in size and pay attention to the font; the images should also be cleaner to make it easier to distinguish the scales
· I suggest that the authors explain what OR(1.22; ….....) means and what the cutoff for OR is or is it arbitrarily set for each cluster analysis and comparison
Author Response
Response to Reviewer 2
PS ECOG under 3 should explain in more detail what this means in order to facilitate monitoring and understanding of respondents' status. “Every patient needs to be fit for oncological treatment with a PS ECOG under 3. Exclusion criteria were: ECOG Performance Status over 2“ this part should be clarified more precisely, because if they are more than 2 as an exclusion factor, and in the previous sentence it is under 3
Authors’ reply: We are grateful to the reviewer for his helpful comments and suggestions and apologise if there was any confusion in this section of the paper. It is undoubtedly a confusing way of describing, which escaped a final check of the paper, that the patients recruited had an ECOG no higher than 2. Following the Reviewer’s criticism the text has been modified.
The authors should clarify how they did it: "We constructed the ANTIBLASTIC DRUG MULTIPANEL PLATFORM?
Authors’ reply: We agree with the reviewer, the sentence is not appropriate. Following the Reviewer’s criticism the text has been modified.
Table 1 - the table should not be split into two pages, what is DS? Mean (SD) ? The authors should explain this with a short explanation below the table
Authors’ reply: Following the Reviewer’s suggestion, the Table 1 has been corrected. In the paper we sent to the journal's editorial office, the table was on a single page and the corresponding legend was placed above the table. It is likely that the table is split on two pages due to the formatting of the paper by the editor. We will point this out in our cover letter to the editor.
Figure 4 - make the diagrams uniform in size and pay attention to the font; the images should also be cleaner to make it easier to distinguish the scales
Authors’ reply: Following the Reviewer’s suggestion, the Figure 4 has been corrected.
I suggest that the authors explain what OR (1.22; ….....) means and what the cutoff for OR is or is it arbitrarily set for each cluster analysis and comparison
Authors’ reply: We apologize for not indicating the lower limit of report Odds ratio that is conventionally greater than 1. We used the Odds ratio for measuring the association between clusters and ADR outcomes. In our study an odds ratio greater than 1 indicates that the event is more likely to occur in the first group, while an odds ratio less than 1 indicates that the condition or event is less likely to occur in the first group. We preferred to use odds ratio instead conventional tests although results greater than 1 would indicate a signal in a two-sided test with a 95% confidence interval (CI). Therefore, we implement the phrase “Thus, we used MedCalc software for OR calculation (OR values with upper and lower limits calculated with a 95% CI) starting from clusters and ADR outcomes for each treatment.” as follows “Thus, we used MedCalc software for OR calculation (OR values with upper and lower limits calculated with a 95% CI) starting from clusters and ADR outcomes for each treatment. The odds ratio equal 1 (the same as a probability of 0.5 or 95% CI) means that exposure does not affect odds of outcome, an Odds >1 implies that exposure is associated with higher odds of outcome; and OR<1 means that exposure is associated with lower odds of outcome.” (page 9 lines 341-343, revised manuscript). Regarding the “what OR (1.22; ….....)”, it means that the odds of a defined cluster is 20% higher in the considered cluster. As well, Odds = 1.40 means 40% more likely, if the odds ratio is 1.50 means a 50% more likely and so on.
Reviewer 3 Report
Comments and Suggestions for Authors
This study presents a pharmacogenetic platform with the potential to transform ADR prediction in NSCLC patients undergoing chemotherapy. While the approach is clinically relevant, I have some concerns as below, addressing these issues could enhance this platform.
1) The study includes only 70 NSCLC patients, which may limit the statistical power and generalizability of the findings. A larger, more diverse cohort would strengthen the validation of the platform.
2) While the platform uses 326 SNPs, the criteria for selecting these specific SNPs are not detailed. Were these SNPs identified from prior studies, or were they based on novel research? This is critical for understanding the platform’s reliability and potential reproducibility.
3) The study mentions comparing therapeutic risks with a “no-treatment scenario,” but it does not clearly describe baseline comparisons or controls. For example, how does this model compare with other predictive methods or standard clinical risk assessments?
4) While clustering patients based on SNPs is innovative, the study does not explain how the clusters were validated or if the clustering process was reproducible. Additional details on cluster stability or cross-validation would improve confidence in the results.
5) There are some other tools (PMID: 39206872) that are able to do similar prediction, it is necessary to illustrate their differences.
Author Response
Response to Reviewer 3
This study presents a pharmacogenetic platform with the potential to transform ADR prediction in NSCLC patients undergoing chemotherapy. While the approach is clinically relevant, I have some concerns as below, addressing these issues could enhance this platform.
Authors’ reply: We would like to thank the Reviewer for reading accurately our work and expressing constructive criticism.
However, we would like to make it clear that our observational study does not pretend to be a clinical study but, as we have also stated in the paper “Our goal was focused on the technical and methodological validation of this patented hybrid platform rather than a clinical validation, for which additional patient cohorts and larger numbers are definitely needed. Obviously, clinical validation requires a different methodological approach, not simply correlational and observational, but also interventional ones. In other words, a validation study in which the patient's treatment is expected to be changed and modified according to the prospective prediction of the response to adverse drug reactions based on the genotype identified in the correlative phase (Discussion. Page 12)”. So this is an “initial,” as also described in the title of the paper, based on a methodological validation, as even a clinical study using an experimental platform would be unlikely to be approved.
Furthermore, we would like to point out that the method we developed is potentially applicable to several malignancies, as we stated in the text “Since our approach includes SNPs related to common cancer therapies, it can be also applied, after appropriate bioinformatic evaluation, to other cancer models” (Discussion, page 11). The choice to analyse NSCLC is purely practical, such as a model to validate our device. Moreover, since for an primary validation it was necessary to have at least a homogeneous population of a certain size in order to proceed with the study, our study population included 70 NSCLC patients undergoing chemotherapy. Finally, the funding available to carry on the validation was limited to this neoplasm.
- The study includes only 70 NSCLC patients, which may limit the statistical power and generalizability of the findings. A larger, more diverse cohort would strengthen the validation of the platform.
Authors’ reply: We agree with the reviewer, and we highlighted this point in the paper “larger numbers are definitely needed”, but we believe that the study achieved our original goal of developing a bioinformatics system that can easily take data from an NGS analysis results and generate a report, also in graphical form, of the likely drug toxicities for each patient. (see Figure 3). We also highlight another reason related to the limited budget that was allocated to this study (during the COVID period), which allowed neither an increase in the number of samples nor an extension of the analyses to other types of neoplasia.
2) While the platform uses 326 SNPs, the criteria for selecting these specific SNPs are not detailed. Were these SNPs identified from prior studies, or were they based on novel research? This is critical for understanding the platform’s reliability and potential reproducibility.
Authors’ reply: The process by which the SNPs were selected using the Pharmgkb platform (https://www.pharmgkb.org/) is described in detail in the 'Materials and methods' section (page 3).
However, we are aware that the platform is quite difficult to use for those who have never used it before. Briefly, the "search engine" allows operator to type the name of the drug to get to the specific page dedicated to that drug. There are several links on this page, such as 'Clinical Annotations', which can display a list of all the genes and their associated SNPs that are associated with a particular drug response. Conversely, it is also possible to evaluate each SNP and derive the association with different drugs. Using the Variant Annotations link, it is also possible to assess the SNP genotype (allele) association with drug response. Of course, this is only part of the functions and data available, all of which are drawn from the extensive scientific reference literature that the site's editors take care to translate into computer language and keep up to date.
3) The study mentions comparing therapeutic risks with a “no-treatment scenario,” but it does not clearly describe baseline comparisons or controls. For example, how does this model compare with other predictive methods or standard clinical risk assessments?
Authors’ reply: We apologise for it, this sentence is completely wrong. We thank the reviewer for pointing this out. Following the reviewer's valuable suggestion, we have edited and rephrased the sentence and moved it one paragraph down where we think it is more logically placed.
4) While clustering patients based on SNPs is innovative, the study does not explain how the clusters were validated or if the clustering process was reproducible. Additional details on cluster stability or cross-validation would improve confidence in the results.
Authors’ reply: First, we made sure that different bioinformatic analyses always provided the same results. Also, as specified in the Results section “we conducted Gene Ontology (GO) enrichment analysis. This analysis classified genes into specific categories related to their “Biological Process” and “Molecular Function” within each cluster of patients, as detailed in Supplementary Figure 2 and Supplementary Table 4. Each patient cluster is characterized by unique combinations of variants, where each variant is associated with distinct biological processes and molecular functions. This analysis demonstrated that each cluster identify patients with uniquely defined biological activities and functional capabilities”.
5) There are some other tools (PMID: 39206872) that are able to do similar prediction, it is necessary to illustrate their differences.
Authors’ reply: The text was implemented following the reviewer's suggestions.
Round 2
Reviewer 3 Report
Comments and Suggestions for Authors
I have no more concerns about this manuscript. It can be accepted for application.